# COMPLETION CONSISTENCY FOR POINT CLOUD COMPLETION ENHANCEMENT

## ABSTRACT

Point cloud completion networks are conventionally trained to minimize the disparities between the completed point cloud and the ground-truth counterpart. However, an incomplete object-level point cloud can have multiple valid completion solutions when it is examined in isolation. This one-to-many mapping issue can cause contradictory supervision signals to the network, because the loss function may produce various values for identical input-output pairs of the network. And in many cases, this issue could adversely impact the network optimization process. In this work, we propose to enhance the conventional learning objective using a novel completion consistency loss to mitigate the one-to-many mapping problem. Specifically, the proposed consistency loss imposes a constraint to ensure that a point cloud completion network generates a consistent completion solution for incomplete objects originating from the same source point cloud. Experimental results across multiple well-established datasets and benchmarks demonstrate the excellent capability of the proposed completion consistency loss to enhance the completion performance of various existing networks without any modification to the design of the networks.

## 1 INTRODUCTION

In recent years, numerous studies (Yang et al., 2018; Tchapmi et al., 2019; Huang et al., 2020; Wen et al., 2021; Chen et al., 2023) have been conducted to leverage deep neural networks to complete occluded object-level point clouds[1]. These point cloud completion networks (PCCNs) are often designed to take locally-incomplete point clouds as input and generate complete point clouds as output. Although recent developments in PCCNs have led to a steady improvement of the completion performance, achieving accurate point cloud completion for a diverse set of objects remains challenging. This challenge is apparent when we compare the completion accuracy of state-of-the-art PCCNs on two different benchmarks: PCN (Yuan et al., 2018) that consists of 30K point clouds from 8 shape categories, and the more diverse Shapenet55 (Yu et al., 2021) benchmark that consists of 52K point clouds from 55 shape categories. Given that objects in the real-world are often diverse, it is important to bridge this gap and improve the completion accuracy of PCCNs for a diverse set of objects.

Improvements of the completion performance of recent PCCNs can primarily be attributed to innovations in network architectures (Yuan et al., 2018; Yu et al., 2021; Zhang et al., 2022), point generation strategies (Xiang et al., 2021; Tang et al., 2022; Wen et al., 2022), or representations (Zhou et al., 2022). On the other hand, the training strategy employed by existing PCCNs has remained relatively unchanged, that is, to minimize the dissimilarities between the predicted complete point clouds and the ground truths (Fei et al., 2022), often measured using the computationally efficient Chamfer Distance metric (Fan et al., 2017). Unfortunately, the straightforwardness of such a training strategy is not without a potential drawback: an incomplete point cloud, when inspected independently without additional information, could have multiple valid solutions according to the CD metric.

To illustrate, consider a simple scenario in which an incomplete point cloud has a partial cuboid shape, shown in Figure 1. This incomplete point cloud can be obtained from various objects such as

---

[1] we use the terms "object" and "point cloud" interchangeably to refer to object-level point clouds

a table, a bed, or other type of objects. We hypothesize that such scenarios can lead to contradictory supervision signals during the training process, in which the loss function could yield various values for the same input-output pairs. As a result, at the end of the training process, the network might produce middle-ground-solutions for both inputs that are suboptimal in terms of completion quality.

In this work, we investigate the effects of these one-to-many mapping cases and obtain evidence that support our hypothesis: the one-to-many mapping problem can negatively affect the training process of PCCNs. To address this, we propose a novel completion consistency loss that can be easily integrated into the commonly-used training strategy, without any changes to the design of the networks. The core idea of the completion consistency loss is to examine multiple incomplete views of a source object at the same time instead of inspecting them independently. That is, at each forward-backward pass, we sample a set of incomplete point clouds originating from the same object, and take a gradient descent step with considerations to the fact that the completion solutions for each element in this set should be identical. This is in contrast to the conventional training strategy, in which only one incomplete point cloud is considered for each source object at each forward-backward pass.

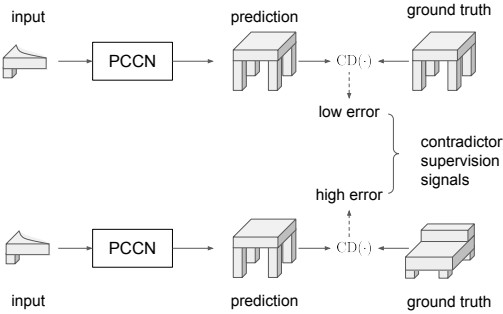

Figure 1: Contradictory supervision signals could appear when an incomplete point cloud have multiple possible completion solutions, and could lead the network to fall into suboptimal solution regions. Point clouds are represented with solid lines in the figure for clarity.

To demonstrate the effectiveness of the completion consistency loss, we evaluate three existing PCCNs, PCN (Yuan et al., 2018), AxFormNet (Zhang et al., 2022), and AdaPoinTr (Yu et al., 2023), on well-established benchmarks (Yuan et al., 2018; Yu et al., 2021), without any modifications to the original network architectures. In all three networks, the completion performance is improved when the completion consistency loss is used during the training. Furthermore, we observe that relatively fast but simple PCCNs (PCN and AxFormNet) that are trained with the consistency loss can match the completion accuracy of more complex but slower PCCNs. In addition, experimental results indicate that the consistency loss can improve the capability of the networks to generalize to previously unseen shape categories. Therefore, the consistency loss could pave the way for accurate, fast, and robust PCCNs, especially for completing a set of point clouds with diverse shapes.

## 2  BACKGROUND

### 2.1  RELATED WORK

Traditional approaches (Wu et al., 2015; Dai et al., 2017; Girdhar et al., 2016; Han et al., 2017) for 3D shape completion task often use voxels as the data representation. However, the memory requirement for voxel-based operations grows cubically with respect to the spatial resolution. In contrast, point cloud representation is capable of preserving 3D structure details with low memory requirement, and has become widely-used in many deep learning applications owing to the pioneering works of Qi et al. (2017a) and Qi et al. (2017b).

PCN (Yuan et al., 2018) is one of the first deep learning-based neural networks for point cloud completion. It utilizes an encoder-decoder-folding scheme to learn features from the partial point cloud, and predicts the final reconstructed points with FoldingNet (Yang et al., 2018). Since then, numerous network architectures for point cloud completion have been proposed. For example, TopNet (Tchapmi et al., 2019) with softly-constrained decoder that is capable of generating point clouds based on a hierarchical rooted tree structure and GRNet (Xie et al., 2020) that leverages gridding operations to enable point cloud to 3D grids transformation without loss of structural information.

Recently, attention-based architectures have grown in popularity as the go-to architecture for PCCN. For example, PoinTr Yu et al. (2021) use a geometry-aware transformer architecture to estimate coarse point predictions before performing refinement via FoldingNet (Yang et al., 2018), while Seedformer (Zhou et al., 2022) introduces Patch Seeds as a new shape representation which contains seed coordinates and features of a small region in the point cloud.

## 2.2 OPTIMAL TRAINING STRATEGY CAN IMPROVE COMPLETION PERFORMANCE

The works discussed in Subsection 2.1 mainly focus on architectural innovations to improve the state-of-the-art point cloud completion performance. On the other hand, several works (Liu et al., 2022; Qian et al., 2022; Steiner et al., 2022) have highlighted that a well-designed training strategy can improve the performance of a neural network. As such, we posit that developing a good training strategy could yield similar advantages for the completion performance of PCCNs.

A training strategy covers a wide array of aspects including the choice of optimizer, learning rate schedule, regularization techniques, data augmentations, auxiliary tasks, and more. To emphasize the significance of a well-designed training strategy, we train a PCN (Yuan et al., 2018) model using the AdamW (Loshchilov & Hutter, 2017) optimizer for 250 epochs, with a cosine annealing (Loshchilov & Hutter, 2016) scheduler. We set the maximum and minimum learning rates to $10^{-4}$ and $5 \cdot 10^{-5}$, respectively, and keep the network architecture and other hyperparameters identical with those used by Yu et al. (2021).

Table 1: Completion performance on ShapeNet55-hard where 75% of the original points are missing. [1]As reported in the ShapeNet55 benchmark (Yu et al., 2021).

| Model | $CD_{l2} \times 10^3 \downarrow$ |
|---|---|
| PCN[1] | 4.08 |
| + Improved Training | 2.37 |
| PoinTr[1] | 1.79 |

As shown in Table 1, the PCN model trained with this improved strategy achieved a $CD_{l2}$ score of $2.37 \cdot 10^{-3}$, a substantial improvement over the previously reported performance of $4.08 \cdot 10^{-3}$, and closer to the completion performance of more recent transformer-based models such as PoinTr (Yu et al., 2021). This result clearly demonstrates the positive impacts of a good training strategy to the completion performance of a PCCN.

## 2.3 LEARNING TO PREDICT ONLY THE MISSING POINTS CAN IMPROVE COMPLETION PERFORMANCE

Another aspect of training strategy for PCCNs is the formulation of the point cloud completion problem. In the literature, there are at least two major problem formulation for deep learning-based point cloud completion. Let $\mathbb{P}^{com}$ be a set of points $p_i^{com} \in \mathbb{R}^3$ sampled from an object $O$ and $\Phi$ be a neural network. We can obtain two disjoint sets from $\mathbb{P}^{com}$: the set of missing points $\mathbb{P}^{mis}$ and the set of incomplete points $\mathbb{P}^{inc}$, where $\mathbb{P}^{com} = \mathbb{P}^{mis} \cup \mathbb{P}^{inc}$ and $\mathbb{P}^{mis} \cap \mathbb{P}^{inc} = \emptyset$.

In the first approach (Yuan et al., 2018; Zhang et al., 2022), the goal is to estimate the entire complete point cloud given an incomplete point cloud, $\Phi(\mathbb{P}^{inc}) = \hat{\mathbb{P}}^{com}$ and minimize the completion error as measured by the Chamfer Distance, $CD(\hat{\mathbb{P}}^{com}, \mathbb{P}^{com})$. In the second approach (Yu et al., 2021), the goal is to estimate only the missing point cloud given an incomplete point cloud, $\Phi(\mathbb{P}^{inc}) = \hat{\mathbb{P}}^{mis}$ and minimize $CD(\hat{\mathbb{P}}^{mis}, \mathbb{P}^{mis})$. The estimated complete point cloud of the second approach is then the union of the predicted missing points and the input incomplete points, $\hat{\mathbb{P}}^{com} = \hat{\mathbb{P}}^{mis} \cup \mathbb{P}^{inc}$.

Table 2: Completion performance on ShapeNet55-hard where 75% of the original points are missing. We use AxForm (Zhang et al., 2022) as $\Phi$.

| Model | $CD_{l2} \times 10^3 \downarrow$ |
|---|---|
| $\Phi(\mathbb{P}^{inc}) = \hat{\mathbb{P}}^{mis}$ | 1.62 |
| $\Phi(\mathbb{P}^{inc}) = \hat{\mathbb{P}}^{com}$ | 1.80 |

To compare the completion performance between the two approaches, we train two AxForm networks (Zhang et al., 2022), one for each approach. As shown in 2, the second approach (predicting

only the missing points) yields better completion performance than the first approach (predicting complete points). Therefore, the experiments in the following sections are based on the second approach, for which the objective can be considered as a reconstruction loss,

$$\mathcal{L}_k^{\text{rec}} = \text{CD}(\hat{\mathbb{P}}_k^{\text{mis}}, \mathbb{P}_k^{\text{mis}}), \tag{1}$$

where CD is defined as,

$$\text{CD}(\mathbb{A}, \mathbb{B}) = \frac{1}{|\mathbb{A}|} \sum_{\boldsymbol{a} \in \mathbb{A}} \min_{\boldsymbol{b} \in \mathbb{B}} ||\boldsymbol{a} - \boldsymbol{b}||_2^2 + \frac{1}{|\mathbb{B}|} \sum_{\boldsymbol{b} \in \mathbb{B}} \min_{\boldsymbol{a} \in \mathbb{A}} ||\boldsymbol{b} - \boldsymbol{a}||_2^2. \tag{2}$$

## 2.4 ONE-TO-MANY MAPPING ISSUE CAN WORSEN THE COMPLETION PERFORMANCE

To investigate the potential impact of the one-to-many mapping issue on the completion performance of PCCNs, we conduct experiments on toy datasets that are derived from the Shapenet55 dataset. First, we construct two types of toy datasets, $\mathbb{D}^A = \bigcup_{i=1}^{5} \mathbb{D}_i^A$ and $\mathbb{D}^B = \bigcup_{i=1}^{5} \mathbb{D}_i^B$, where $\mathbb{D}_i^A$ and $\mathbb{D}_i^B$ each consists of 5,000 samples from ShapeNet55. The samples in $\mathbb{D}_i^A$ is selected in a way such that, on average, the CD-score between $\mathbb{P}_j^{\text{inc}} \in \mathbb{D}_i^A$ and $\mathbb{P}_k^{\text{inc}} \in \mathbb{D}_i^A$, $j \neq k$, is relatively low, but the CD-

Table 3: Completion performance on Toy Datasets based on ShapeNet55-hard.

| Model | $\text{CD}_{l2} \times 10^3 \downarrow$ |
|---|---|
| AxForm on $\mathbb{D}^A$ | $2.81 \pm 0.15$ |
| AxForm on $\mathbb{D}^B$ | $2.44 \pm 0.10$ |

score between $\mathbb{P}_j^{\text{mis}} \in \mathbb{D}_i^A$ and $\mathbb{P}_k^{\text{mis}} \in \mathbb{D}_i^A$, $j \neq k$, is relatively high. Further details regarding the steps to generate $\mathbb{D}^A$ can be found in Appendix A.1. Meanwhile, samples in $\mathbb{D}_i^B$ are randomly selected from Shapenet55 with uniform probabilities and therefore is statistically similar to the full ShapeNet55 dataset.

We use 80% of the samples in each dataset for training, and hold the remaining 20% for evaluation. In total, we train 10 AxForm networks (Zhang et al., 2022) on $\mathbb{D}^A$ and $\mathbb{D}^B$, and report the average and standard deviation of the CD-scores. As shown in Table 3, the CD-score of networks trained and evaluated on $\mathbb{D}^B$ is lower (better) than the CD-score of networks trained and evaluated on $\mathbb{D}^A$. These results indicate that the one-to-many mapping issue negatively affects the completion performance of the PCCNs.

## 3 MAINTAINING COMPLETION CONSISTENCY

In this section, we introduce the completion consistency loss, which we refer to as the consistency loss for brevity from here onward, to mitigate the aforementioned issues. The core idea of the consistency loss is to consider multiple incomplete point clouds originating from the same source object before taking a gradient descent step (Figure 2). Recall that the contradictory supervision signals exist when there are multiple valid completion solutions for one incomplete point cloud that is observed in isolation. Therefore, intuitively, adding more incomplete point clouds with the same completion solution at one observation can reduce the ambiguity and mitigate the negative effects of the issue.

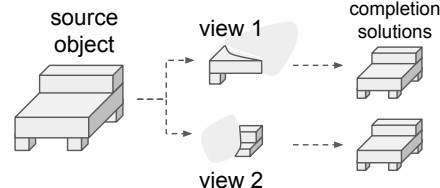

Figure 2: Two different incomplete point clouds that are obtained from one object should have the same solutions.

## 3.1 COMPLETION CONSISTENCY LOSS

We propose two ways to implement the consistency loss: self-guided consistency and target-guided consistency.

**Self-guided Consistency.** In self-guided consistency loss, we leverage the fact that we can generate multiple incomplete point clouds from the same object, and utilize these samples in the consistency

loss. Given a complete point cloud $\mathbb{P}_k^{\text{com}}$ representing the object $k$, we can generate a set of $n$ different incomplete point clouds $\mathbb{P}_k^{\text{inc}} = \{\mathbb{P}_{k,1}^{\text{inc}}, \mathbb{P}_{k,2}^{\text{inc}}, ..., \mathbb{P}_{k,n}^{\text{inc}}\}$. Since the source of all incomplete point clouds is the same, that is, $\mathbb{P}_k^{\text{com}}$, the completion solutions for all $\mathbb{P}_{k,i}^{\text{inc}}$ should also be the same. Therefore, given $\Phi(\mathbb{P}_{k,i}^{\text{inc}}) = \hat{\mathbb{P}}_{k,i}^{\text{mis}}$ and $\hat{\mathbb{P}}_{k,i}^{\text{com}} = \hat{\mathbb{P}}_{k,i}^{\text{mis}} \cup \mathbb{P}_{k,i}^{\text{inc}}$, we can guide the network to produce similar completion solutions for any incomplete point clouds originating from $\mathbb{P}_k^{\text{com}}$ through the self-guided consistency,

$$\mathcal{L}_k^{\text{c-sg}} = \frac{2}{n(n-1)} \sum_{i=1}^{n-1} \sum_{j=i+1}^{n} \text{CD}(\hat{\mathbb{P}}_{k,i}^{\text{com}}, \hat{\mathbb{P}}_{k,j}^{\text{com}})$$

**Target-guided Consistency.** For target-guided consistency, we utilize the original ground truth for the consistency loss. As mentioned in Subsection 2.3, the commonly-used loss function is calculated as either $\text{CD}(\Phi(\mathbb{P}^{\text{inc}}), \mathbb{P}^{\text{com}})$ or $\text{CD}(\Phi(\mathbb{P}^{\text{inc}}), \mathbb{P}^{\text{mis}})$. While the latter formulation is found improve the completion performance of PCCNs, it does not promote consistency between completions because the supervision is only performed on $\mathbb{P}^{\text{mis}}$ instead of $\mathbb{P}^{\text{com}}$. In target-guided consistency, we propose to keep the approach of predicting only the missing points, but we calculate the loss values based on the full complete point clouds. Specifically, given a complete point cloud $\mathbb{P}_k^{\text{com}}$, $\Phi(\mathbb{P}_{k,i}^{\text{inc}}) = \hat{\mathbb{P}}_{k,i}^{\text{mis}}$ and $\hat{\mathbb{P}}_{k,i}^{\text{com}} = \hat{\mathbb{P}}_{k,i}^{\text{mis}} \cup \mathbb{P}_{k,i}^{\text{inc}}$, the target-guided consistency is defined as,

$$\mathcal{L}_k^{\text{c-tg}} = \frac{1}{n} \sum_{i=1}^{n} \text{CD}(\hat{\mathbb{P}}_{k,i}^{\text{com}}, \mathbb{P}_k^{\text{com}})$$

**Complete Loss Function.** The complete loss function for a complete point cloud $\mathbb{P}_k^{\text{com}}$ with $n$ samples of incomplete point clouds is the combination of conventional reconstruction loss, self-guided consistency loss, and target-guided consistency loss, with scaling factors $\alpha$ and $\beta$,

$$\mathcal{L}_k^{\text{total}} = \alpha \mathcal{L}_k^{\text{c-sg}} + \beta \mathcal{L}_k^{\text{c-tg}} + \frac{1}{n} \sum_{i=1}^{n} \mathcal{L}_{i,k}^{\text{rec}}, \tag{3}$$

where $\mathcal{L}_{i,k}^{\text{rec}}$ is the reconstruction loss (Equation 1) for $\hat{\mathbb{P}}_{k,i}^{\text{mis}}$.

We note that both consistency losses do not directly eliminate the one-to-many mapping issue, but they can provide the network with additional information such that the network can mitigate the issue. For a simple example, consider two inputs $\mathbb{P}_{a,1}^{\text{inc}}$ and $\mathbb{P}_{b,1}^{\text{inc}}$, and the corresponding completion solutions $\mathbb{P}_a^{\text{com}}$ and $\mathbb{P}_b^{\text{com}}$. Suppose that $\text{CD}(\mathbb{P}_{a,1}^{\text{inc}}, \mathbb{P}_{b,1}^{\text{inc}}) \approx 0$, and $\text{CD}(\mathbb{P}_a^{\text{com}}, \mathbb{P}_b^{\text{com}}) >> \text{CD}(\mathbb{P}_{a,1}^{\text{inc}}, \mathbb{P}_{b,1}^{\text{inc}})$, that is, the inputs are similar but the ground truths are dissimilar. Assuming that $\Phi(\mathbb{P}_{a,1}^{\text{inc}})$ is also similar to $\Phi(\mathbb{P}_{b,1}^{\text{inc}})$, then a contradictory supervision signal could arise when we only use $\mathcal{L}^{\text{rec}}$ as the loss function. On the other hand, suppose that we supplement the loss function with the consistency loss with $n = 3$ such that the inputs become $\{\mathbb{P}_{a,1}^{\text{inc}}, \mathbb{P}_{a,2}^{\text{inc}}, \mathbb{P}_{a,3}^{\text{inc}}\}$ and $\{\mathbb{P}_{b,1}^{\text{inc}}, \mathbb{P}_{b,2}^{\text{inc}}, \mathbb{P}_{b,3}^{\text{inc}}\}$ for each ground truth. The effect of the contradictory supervision signal to the gradient descent step can then be suppressed by $\mathcal{L}_k^{\text{c-sg}}$ and $\mathcal{L}_k^{\text{c-tg}}$.

## 4 EXPERIMENTAL RESULTS

In this section, we demonstrate the effectiveness of the consistency loss by comparing the completion performance of three existing PCCNs on three commonly-used datasets. First, we explain the experimental setups that are needed to reproduce the results. Then, we report and discuss the completion performance of three existing PCCNs trained with and without the consistency loss. We also conduct additional experiments to check the effects of each component in the consistency loss.

### 4.1 EXPERIMENTAL SETUP

#### 4.1.1 DATASETS

There are numerous object-level point clouds datasets, most of which are derived from the Shapenet dataset Chang et al. (2015), for example, PCN Yuan et al. (2018), Completion3D Tchapmi et al. (2019), and Shapenet55-34 Yu et al. (2021). We choose to evaluate the consistency loss on the PCN and Shapenet55-34 datasets, following Yu et al. (2021); Zhou et al. (2022); Yu et al. (2023).

PCN consists of around 30K samples of point clouds, spanning over 8 categories: airplane, cabinet, car, chair, lamp, sofa, table, and vessel. On the other hand, Shapenet55-34 consists of around 52K samples of point clouds from 55 categories, resulting in a considerably more diverse set of objects compared with PCN. In Shapenet55, the dataset is split into 41,952 samples for training and 10,518 samples for evaluation, with samples from all 55 categories are present in both training and evaluation splits. Meanwhile in Shapenet34, the dataset is split into 46,765 samples for training and 5,705 samples for evaluation, where the training split consists of samples from 34 categories, and the evaluation split consists of samples from all 55 categories. Shapenet34 can be seen as an evaluation on out-of-distribution data since the 21 extra categories on the evaluation split are withheld during training.

#### 4.1.2 IMPLEMENTATION DETAILS

The consistency loss is designed to improve a PCCN without any modification to the architecture of the network. Therefore, we choose to evaluate the effectiveness of the consistency loss using three existing PCCNs, PCN (Yuan et al., 2018), AxFormNet (Zhang et al., 2022), and AdaPoinTr (Yu et al., 2023). For fairness, we train two versions of all three PCCNs from scratch using publicly-available source codes and the same training strategy, e.g., identical problem formulation, optimizer, number of iterations, batch size, and learning rate schedule. The only difference between the two versions is that whether the consistency loss is incorporated into the loss function or not.

All PCCNs are implemented with PyTorch (Paszke et al., 2019) and trained on RTX 3090 GPUs. The batch sizes are set to 64, 64, and 16 for PCN, AxFormNet, and AdaPoinTr, respectively. We set the number of epochs to 200, 400, and 600 for PCN, AxFormNet, and AdaPoinTr, respectively, utilize cosine annealing (Loshchilov & Hutter, 2016) for the learning rate schedule, and set $n = 3$ for the consistency loss. We use Open3D (Zhou et al., 2018) to visualize the point clouds.

### 4.2 MAIN RESULTS

#### 4.2.1 QUANTITATIVE RESULTS

Following Yu et al. (2021), we report the $CD_{l2}$ metric on three difficulty levels for Shapenet55 and the $CD_l1$ metric for PCN in Table 4. From the results, we can draw the following conclusions,

**The consistency loss consistently improves the completion performance of the three PCCNs.** As shown in Table 4, the consistency loss significantly improves the completion performance of PCN, AxFormNet, and AdaPoinTr on Shapenet55 that consists of objects with diverse geometrical shapes. Specifically, the completion performance is improved by 27%, 25%, and 4.8% for PCN, AxFormNet, and AdaPoinTr, respectively. Similar improvements can also be seen on Shapenet34 (Table 5), in which the mean CD of all three PCCNs trained with the consistency loss are lower or equal to the mean CD of their original counterparts. Additionally, the consistency loss can, to some extent, improve the completion performance of the PCCNs in datasets with less diversity such as PCN. These results demonstrate the effectiveness of the consistency loss for improving the completion performance of existing PCCNs, especially when we are interested in completing a collection of point clouds with diverse geometrical shapes.

**The consistency loss enables fast and accurate point cloud completion.** Point cloud completion is often used as an auxilliary task, therefore, the completion process should be fast to avoid unnecessary overhead to the overall process. However, recent PCCNs such as PoinTr (Yu et al., 2021) and SeedFormer (Zhou et al., 2022) achieve improved completion performance at the cost of inference latency due to the complex design of the network.

Table 4: Quantitative results on the ShapeNet55 (Yu et al., 2021) and PCN (Yuan et al., 2018) benchmarks. We report the L2-norm Chamfer Distance ($CD_{l2}$) and L1-norm Chamfer Disctance ($CD_{l1}$) for ShapeNet55 and PCN, respectively. S, M, and H represent the simple, moderate, and hard setups, where the proportions of missing points are 25%, 50%, and 75%, respectively. * and † indicates that the models are trained from scratch based on source codes from Yu et al. (2021) and Zhang et al. (2022), respectively.

| | ShapeNet55 | | | | PCN |
| | S | M | H | Avg. | Avg. |
| | | $CD_{l2} \times 10^3 \downarrow$ | | | $CD_{l1} \times 10^3 \downarrow$ |
|---|---|---|---|---|---|
| FoldingNet (Yang et al., 2018) | 2.67 | 2.66 | 4.05 | 3.12 | 14.31 |
| PCN* (Yuan et al., 2018) | 0.82 | 1.25 | 2.37 | 1.48 | 10.55 |
|   + Consistency Loss | **0.54** | **0.93** | **1.74** | **1.07** | **10.52** |
| TopNet (Tchapmi et al., 2019) | 2.26 | 2.16 | 4.30 | 2.91 | 12.15 |
| GRNet (Xie et al., 2020) | 1.35 | 1.71 | 2.85 | 1.97 | 8.83 |
| SnowflakeNet (Wen et al., 2021) | 0.70 | 1.06 | 1.96 | 1.24 | 7.21 |
| PoinTr (Yu et al., 2021) | 0.58 | 0.88 | 1.79 | 1.09 | 8.38 |
| AXFormNet† (Zhang et al., 2022) | 0.72 | 1.06 | 1.98 | 1.22 | |
|   + Consistency Loss | **0.45** | **0.79** | **1.51** | **0.91** | |
| SeedFormer (Zhou et al., 2022) | 0.50 | 0.77 | 1.49 | 0.92 | 6.74 |
| AdaPoinTr* (Yu et al., 2023) | 0.51 | 0.71 | 1.28 | 0.83 | 6.53 |
|   + Consistency Loss | **0.47** | **0.68** | **1.24** | **0.79** | **6.51** |

Table 5: Quantitative results on the ShapeNet34 benchmark. We report the L2-norm Chamfer Distance ($CD_{l2}$). S, M, and H represent the simple, moderate, and hard setups, where the proportions of missing points are 25%, 50%, and 75%, respectively. $\Delta$ is the gap between the mean CDs of the 21 unseen categories and the 34 seen categories.

| | 34 seen categories | | | | 21 unseen categories | | | | $\Delta$ |
| | S | M | H | Avg. | S | M | H | Avg. | |
| | | | | $CD_{l2} \times 10^3 \downarrow$ | | | | | |
|---|---|---|---|---|---|---|---|---|---|
| FoldingNet | 1.86 | 1.81 | 3.38 | 2.35 | 2.76 | 2.74 | 5.36 | 3.62 | |
| PCN | 0.84 | 1.26 | 2.37 | 1.49 | 1.41 | 2.28 | 4.63 | 2.77 | 1.28 |
|   + Consistency Loss | **0.57** | **0.96** | **1.76** | **1.09** | **1.07** | **1.84** | **3.70** | **2.20** | **1.11** |
| TopNet | 1.77 | 1.61 | 3.54 | 2.31 | 2.62 | 2.43 | 5.44 | 3.50 | |
| GRNet | 1.26 | 1.39 | 2.57 | 1.74 | 1.85 | 2.25 | 4.87 | 2.99 | |
| SnowflakeNet | 0.60 | 0.86 | 1.50 | 0.99 | 0.88 | 1.46 | 2.92 | 1.75 | |
| PoinTr | 0.76 | 1.05 | 1.88 | 1.23 | 1.04 | 1.67 | 3.44 | 2.05 | |
| AXFormNet | 0.76 | 1.14 | 2.11 | 1.33 | 1.30 | 2.06 | 4.36 | 2.57 | 1.24 |
|   + Consistency Loss | **0.48** | **0.84** | **1.57** | **0.96** | **0.92** | **1.67** | **3.50** | **2.03** | **1.07** |
| SeedFormer | 0.48 | 0.70 | 1.30 | 0.83 | 0.61 | 1.07 | 2.35 | 1.34 | |
| AdaPoinTr | 0.51 | 0.68 | 1.09 | 0.76 | 0.63 | 1.06 | 2.23 | 1.30 | 0.54 |
|   + Consistency Loss | **0.46** | **0.62** | **1.09** | **0.72** | **0.63** | **1.03** | **2.25** | **1.30** | **0.58** |

On the other hand, the proposed consistency loss enables simpler networks to be as accurate as more complex networks, thus improving the completion performance without sacrificing inference latency. Specifically on the Shapenet55 dataset, PCN with consistency loss achieves a mean CD of $1.07 \cdot 10^{-3}$, which is better than the mean CD of PoinTr ($1.09 \cdot 10^{-3}$). Another example is the AxFormNet with consistency loss that achieves a mean CD of $0.91 \cdot 10^{-3}$, which is better than the mean CD of SeedFormer ($0.92 \cdot 10^{-3}$). Considering that, when evaluated on a single RTX 3080Ti GPU, the inference latency of PCN (1.9 ms) and AxFormNet (5.3 ms) are significantly lower than PoinTr (11.8 ms) and SeedFormer (38.3 ms), the consistency loss is a promising training strategy that can enable fast and accurate point cloud completion.

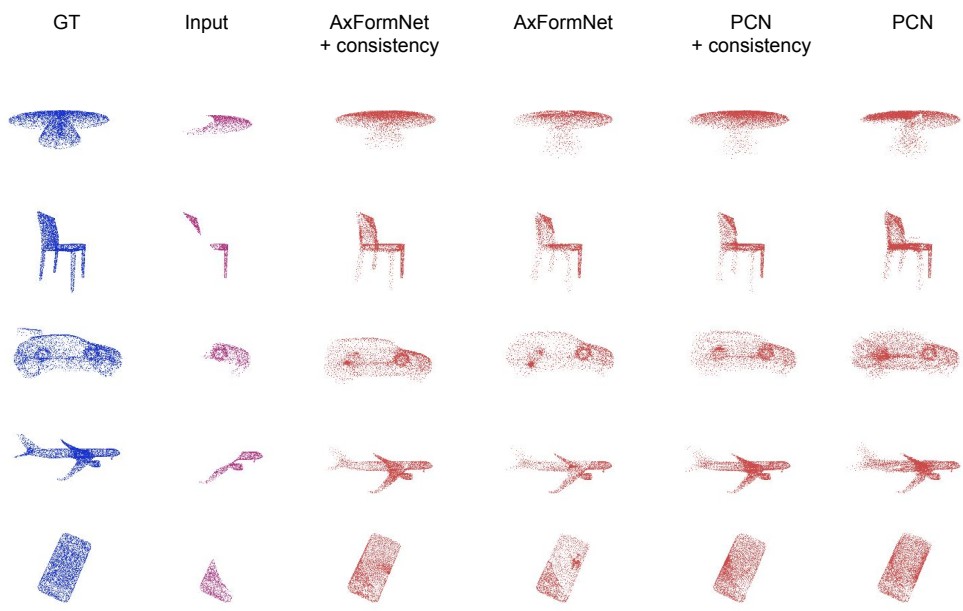

| GT | Input | AxFormNet + consistency | AxFormNet | PCN + consistency | PCN |

Figure 3: Compltetion results on the Shapenet55 dataset (*test* split).

**The consistency loss could improve the generalization capability of PCCNs to previously-unseen objects.** It is desirable for a PCCN to produce accurate completed point clouds even for objects from categories that are not available during training. To quantify the generalization capability of a PCCN, we can consider the gap between the evaluation results on Shapenet34-*seen* split and Shapenet34-*unseen* split, which we refer to as $\Delta$ in Table 5 From the table we can see that incorporating the consistency loss results in a significant improvements in the gaps for PCN and AxFormNet, while the gap for AdaPoinTr stays relatively similar. These results indicate that the consistency loss can act as an additional regularizer for point cloud completion.

### 4.2.2 QUALITATIVE RESULTS ON SHAPENET55 AND SHAPENET34

We visualize the completion results of AxFormNet and PCN on point clouds from the Shapenet55-test and the Shapenet34-*unseen* splits in Figure 3 and Figure 4, respectively. For each object, we use 25% of the points in the point cloud as inputs, which is equivalent to the *hard* setup in Yu et al. (2021). As shown in the figures, networks that are trained with the consistency loss (AxForm-Net+con and PCN+con) predict completed point clouds with equal or better quality compared to the networks that are trained without the consistency loss. For example, on row 1 in Figure 3, AxForm-Net+con can predict the surface of a table with more consistent point density with respect to the ground truth compared to AxFormNet. And PCN+con can predict the complete surface of a table, whereas the surface of a table predicted by PCN contains a missing part on the left side.

### 4.3 ADDITIONAL RESULTS

In the following subsection we show additional results from experiments with AxFormNet to further investigate the effects of the consistency loss. We limit the scope of the experiments to the hardest setup of ShapeNet55 during training and evaluation.

**Number of Training Samples.** To implement the consistency loss, we sample *n* instances of incomplete point clouds per object to be fed to the PCCN. This means that the network has access to *n* times more number of samples during training. A natural question would raise: is the completion accuracy gain simply a result of more training data? To answer this question, we train the original AxFormNet on Shapenet55 with extra budgets, that is, increasing the number of training epochs to

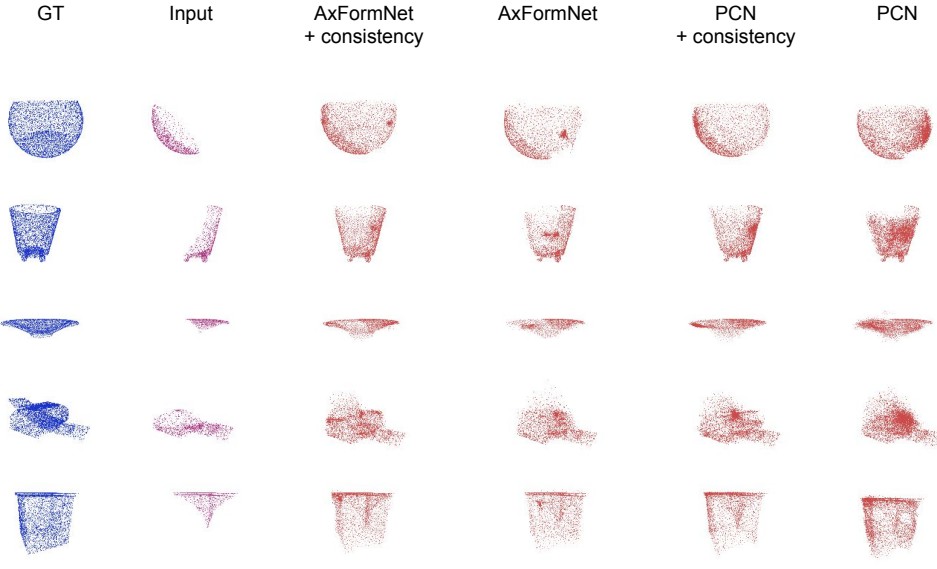

Figure 4: Completion results on the Shapenet34 dataset (*unseen* split).

1200, a threefold increase. We find that the original AxFormNet trained with extra budgets achieves a $CD_{l2} \times 10^3$ score of 1.60, which is worse than AxFormNet trained with the consistency loss ($CD_{l2} \times 10^3 = 1.48$). This result indicates that the completion performance gains in networks trained with the consistency loss are not simply the results of more training data.

**Scaling Factors for $\mathcal{L}^{\text{c-sg}}$ and $\mathcal{L}^{\text{c-tg}}$.** We also investigate the effect of scaling factors $\alpha$ and $\beta$ in Equation 3. As a baseline, we use the AxFormNet network trained to predict the missing point clouds as in Table 2, this is equivalent to $\alpha = \beta = 0$. First, we investigate the individual effect of each component in the consistency loss. From the table we can see that both $\mathcal{L}^{\text{c-tg}}$ ($\beta = 1$) and $\mathcal{L}^{\text{c-sg}}$ ($\alpha = 1$) improve the completion accuracy, with $\mathcal{L}^{\text{c-tg}}$ bringing more benefits compared with $\mathcal{L}^{\text{c-sg}}$. However, when both are used with the same scaling factors (i.e., $\alpha = \beta = 1$), the completion accuracy is worse than when only $\mathcal{L}^{\text{c-tg}}$ is used. From experimental results, we see that setting $\alpha = 0.1$ and $\beta = 1$ yield the best completion accuracy.

Table 6: Completion performance of various AxFormNet on ShapeNet55-hard where 75% of the original points are missing.

| $\mathcal{L}^{\text{c-sg}}(\alpha)$ | $\mathcal{L}^{\text{c-tg}}(\beta)$ | $CD_{l2} \cdot 10^3$ |
|---|---|---|
| 0 | 0 | 1.62 |
| 0 | 1 | 1.51 |
| 1 | 0 | 1.60 |
| 1 | 1 | 1.54 |
| 0.1 | 1 | 1.48 |

## 5 CONCLUSION

We have proposed the completion consistency loss, a novel loss function for point cloud completion. The completion consistency loss has been designed to reduce the adverse effects of contradictory supervision signals by considering multiple incomplete views of a single object in one forward-backward pass. We have demonstrated that the completion consistency loss can improve the completion performance and generalization capability of existing point cloud completion networks without any modification to the design of the networks. Moreover, simple and fast point cloud completion networks that have been trained with the proposed loss function can achieve completion performance similar to more complex and slower networks. Therefore, the completion consistency loss can pave the way for accurate, fast, and robust point cloud completion networks.

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

# A  APPENDIX

## A.1  GENERATING TOY DATASETS

The toy datasets that are used in Subsection 2.4 are generated by following Algorithm 1. CD is the chamfer distance function defined in Equation 2.

---
**Algorithm 1** Generating Toy Datasets
---
**Input**: Full dataset $\mathbb{D}$
Initialize $\mathbb{D}^A$ as an empty tensor, $k_1 \leftarrow 100$, $k_2 \leftarrow 5$, $n \leftarrow 5000$
**while** len$(\mathbb{D}^A) \leq n$ **do**
    Sample $\boldsymbol{X}$ from $\mathbb{D}$
    Initialize $\mathbb{D}^{\text{inc}}$, $\mathbb{D}^{\text{mis}}$, $\mathbb{D}^{\text{inc}}$, $\mathbb{D}^{\text{mis}}$ as empty tensors.
    **for** $\boldsymbol{Y}$ in $\mathbb{D}$ **do**
        Append CD$(\boldsymbol{X}^{\text{inc}}, \boldsymbol{Y}^{\text{inc}})$ to $\mathbb{D}^{\text{inc}}$
    **end for**
    Calculate $k_1$-lowest CD-metric in $\mathbb{D}^{\text{inc}}$
    Append the $k_1$ corresponding $\boldsymbol{Y} \in \mathbb{D}$ to $\mathbb{D}^{\text{inc}}$
    **for** $\boldsymbol{Z}$ in $\mathbb{D}^{\text{inc}}$ **do**
        Append CD$(\boldsymbol{X}^{\text{mis}}, \boldsymbol{Z}^{\text{mis}})$ to $\mathbb{D}^{\text{mis}}$
    **end for**
    Calculate $k_2$-highest CD-metric in $\mathbb{D}^{\text{mis}}$
    Append the $k_2$ corresponding $\boldsymbol{Z} \in \mathbb{D}$ to $\mathbb{D}^{mis}$
    **if** $\boldsymbol{X} \notin \mathbb{D}^A$ **then**
        Append $\boldsymbol{X} \in \mathbb{D}$ to $\mathbb{D}^A$
    **end if**
    **for** $\boldsymbol{Z}$ in $\mathbb{D}^{\text{mis}}$ **do**
        **if** $\boldsymbol{Z} \notin \mathbb{D}^A$ **then**
            Append $\boldsymbol{Z}$ to $\mathbb{D}^A$
        **end if**
    **end for**
**end while**
Return the first $n$ elements in $\mathbb{D}^A$

---

