# OpenReview forum: "Completion Consistency for Point Cloud Completion Enhancement"
_ICLR.cc/2024/Conference — Submitted to ICLR 2024_

### Official Review · Reviewer_uAuJ · 2023-10-28

**Soundness:** 2 fair
**Presentation:** 3 good
**Contribution:** 2 fair
**Rating:** 5
**Confidence:** 3

**Summary:**

The paper focuses on the task of point cloud completion (PCC). Due to the ill-posed nature of CD distance, the current training signal for PCC could be noisy and contradictory, which hinders the resulting completion quality.

The authors thus propose a consistency loss such that a many-to-many loss calculation is established. For each prediction, it is evaluated against a set of ground truth complete point clouds; and for each ground truth complete point cloud, it is also evaluated with a set of predictions. The consistency among these computed losses is encouraged.

**Strengths:**

This paper is well-written.

The proposed method is simple and is demonstrated to be effective to some extent.

**Weaknesses:**

Regarding the analysis of the consistency loss at the end of Sec.3, I think it is not clear and could be introduced in more detail. Specifically, the authors claim that "The effect of the contradictory supervision signal to the gradient descent step can then be suppressed by $L_{k}^{c-{tg}}$ and $L_{k}^{c-{s g}}$", while I think the two losses $L_{k}^{c-{tg}}$ and $L_{k}^{c-{s g}}$ might provide very close constraints. And the results in Tab. 6 could also prove that, such as Row#2 v.s. Row#5 and Row#3 v.s. Row#4. We can observe that the weight of $L_{k}^{c-{s g}}$ should be small, if $L_{k}^{c-{tg}}$ and $L_{k}^{c-{s g}}$ have the same weight, the performance even drops. I think more explanations are required. Otherwise, the motivation behind the solution is not clear. How the method suppresses such signals in practice should be analyzed more, such as from the perspective of gradients.

**Questions:**

- Please give more analysis about the consistency loss, such as from the perspective of gradients.

- Please provide more analysis about the performance comparisons in Tab. 6.

- Please explain the differences and similarities between the two losses $L_{k}^{c-{tg}}$ and $L_{k}^{c-{s g}}$.

---

### Official Review · Reviewer_CxXc · 2023-10-30

**Soundness:** 3 good
**Presentation:** 3 good
**Contribution:** 2 fair
**Rating:** 3
**Confidence:** 5

**Summary:**

The paper addresses the challenge of one-to-many mapping in point cloud completion networks, where an incomplete point cloud can have multiple valid completion solutions. This issue can lead to contradictory supervision signals during training, potentially hindering network optimization. To tackle this, the authors propose a novel completion consistency loss that enforces the generation of a consistent completion solution for incomplete objects from the same source point cloud. Experimental results across different datasets and benchmarks showcase the effectiveness of this approach in improving the completion performance of existing networks without requiring modifications to their design.

**Strengths:**

1. The paper is well organized.
2. The motivation of the paper is important and interesting.

**Weaknesses:**

1. In Tables 4 and 5, the proposed loss functions are plugged into several methods but not all of them. Is there a performance gain for the rest of the methods?
2. Does the marginal performance improvement of AdaPointTr imply that the current state-of-the-art models have somehow implicitly learned to handle ambiguity in supervised signals, proving the proposed losses unnecessary?
3. The primary contribution of the paper lies in the proposed loss functions, which have not been conclusively demonstrated to be highly effective. Therefore, it is believed that the paper has not met the acceptance bar of ICLR.

**Questions:**

Please refer to the weaknesses.

---

### Official Review · Reviewer_MC7Y · 2023-10-30

**Soundness:** 2 fair
**Presentation:** 3 good
**Contribution:** 2 fair
**Rating:** 3
**Confidence:** 5

**Summary:**

The paper proposes a method for point cloud completion using a novel completion consistency loss to mitigate the issue that one-to-many mapping can cause contradictory supervision signals for point cloud completion training. More specifically, the authors propose a self-guided consistency and a target-guided consistency to ensure that the network generates a consistent completion for incomplete objects. To verify the effectiveness, extensive experiments are conducted, and promising results are achieved.

**Strengths:**

1. The paper is clearly written and well organized.
2. Extensive experiments are tested together with promising performances.

**Weaknesses:**

1. It is highly encouraged that the experiments should cover the widely used scenario to predict the whole completion points instead of only predicting the missing parts, since in most cases, the input parietal points itself could be quite noisy and inaccurate. Hence, predicting the whole point sets can provide more accurate and evenly distributed outputs.
2. The paper aims to solve the one-to-many issue, but it seems that self-guided consistency is designed as a many-to-one mechanism. Hence, how does the many-to-one approach help the one-to-many issue? Moreover, self-guided consistency seems not novel, and it has been studied in [A] before.
3. It seems that a SOTA method [B] in ShapeNet55 and ShapeNet34 is missing. Hence, it would be great if authors could provide experimental results based on [B] as well to show the effectiveness of the method.

[A] Gu, Jiayuan, et al. "Weakly-supervised 3D shape completion in the wild." Computer Vision–ECCV 2020: 16th European Conference, Glasgow, UK, August 23–28, 2020, Proceedings, Part V 16. Springer International Publishing, 2020.
[B]Chen, Zhikai, et al. "AnchorFormer: Point Cloud Completion From Discriminative Nodes." Proceedings of the IEEE/CVF Conference on Computer Vision and Pattern Recognition. 2023.

**Questions:**

1. How are the results in Table 2 obtained? Are the CD values calculated from the whole point set (both partial and missing parts)? If so, it is obvious that predicting the missing points alone is better than the other one, since there is always a subset of points that has zero CDs. It would be better to also show the qualitative comparisons to support the authors’ assumption.
2. Why the PCN result (10.55) in Table 4 is different from the original paper’s (9.636)?
3. Why is AXFormNet not tested on the PCN dataset in Table 4?

---

### Official Review · Reviewer_3ARx · 2023-10-31

**Soundness:** 1 poor
**Presentation:** 2 fair
**Contribution:** 1 poor
**Rating:** 3
**Confidence:** 5

**Summary:**

This paper presents a consistency constraint to address the one-to-many mapping issue in the point cloud completion task. In order to mitigate this problem, the authors propose a consistency loss that consists of a Self-guided Consistency loss and a Target-guided Consistency loss, which serve as guiding factors during the training process. However, the novelty of the proposed method is limited, as the authors merely observe the one-to-many mapping issue without providing a solution. Furthermore, the logic and coherence of the manuscript are lacking, and the arrangement of the content is unreasonable. In the experimental section, although the combination of the proposed consistency constraint with existing methods leads to improved performance, there are issues with the experimental settings and evaluation metrics.

**Strengths:**

I can't see obvious strengths.

**Weaknesses:**

1)	The novelty is limited.

2)	The arrangement of the manuscript is unreasonable.

3)	Some claims are casual and cannot be proved.

4)	The experiments are problematic and insufficient to support the proposed method's efficiency. And some significant experiments are missing.

**Questions:**

1)	The novelty is limited. The main contribution of the proposed method is concentrated on the items of Eq. (3); however, the target-guided consistency has been proposed in previous work SeedFormer [1] (see Eq. (7)).

2)	In Section 2.3, the authors argue that learning to predict only the missing points can improve completion performance and validate this hypothesis using a single model, i.e., AxForm. However, the results from a single model are insufficient to establish the validity of this inference. Further experimentation with more models is required to support this claim.

3)	The consistency loss requires the utilization of n incomplete point clouds for each object. However, this manuscript does not mention how these incomplete point clouds are generated. In datasets like PCN, each object is associated with 8 partial point clouds. Do these partial observations overlap with the incomplete point clouds used for the consistency loss?

4)	There is no evidence and theoretical analysis about how the proposed self-guided consistency could solve the one-to-many mapping issue, which is mentioned in Sec.1. The proposed method seems to deal with the many-to-one issue. It is recommended to provide corresponding experiments and results to support this claim.

5)	The arrangement of the manuscript is unreasonable. For example, Sec 2.3 should introduce the related works rather than the experiments. Sec 2.3 is suggested to be put in Sec. 1. The motivation of Sec 2.2 is not clear. The proposed method is about loss function instead of the optimizer in training, but Sec 2.2 introduces and carries the experiments about the optimizer.

6)	Some claims cannot be proved with enough evidence. In the last paragraph in Sec “Complete Loss Function”, why CD(P_{a}^{com}, P_{b}^{com})>>CD(P_{a1}^{inc}, P_{b1}^{inc}) since the CD calculates the average error between point clouds?

7)	The evaluation metrics are problematic. The evaluation metric should include F-Score besides CD.

8)	Evaluating the performance on real-world datasets is crucial to validate the effectiveness of point cloud completion methods. While this paper demonstrates the effectiveness of the proposed method on synthetic datasets, it does not extend the evaluation to real-world datasets such as KITTI.

9)	When combined with the newest frameworks, the increment seems to be unobvious. Does it illustrate that the proposed method is not necessary in these frameworks?

10)	There is no ablation study.

11)	In Fig.3, the improvement in the “chair” is not obvious.

12)	There is no time and GPU memory analysis.

[1] SeedFormer: Patch Seeds Based Point Cloud Completion with Upsample Transformer.

---

### Meta-Review · Area_Chair_s6a6 · 2023-12-05

**Metareview:**

Dear Authors,

Majority of the reviewers have indicated that this draft is not ready for the publication at this stage, and authors have not provided any rebuttal against those concerns.


regards
AC

**Justification For Why Not Higher Score:**

Majority of the reviewers have indicated that this draft is not ready for the publication at this stage, and authors have not provided any rebuttal against those concerns.

**Justification For Why Not Lower Score:**

N/A

---

### Decision · Program_Chairs · 2024-01-16

Reject